# Identification and Characterization of HIRIP3 as a Histone H2A Chaperone

**DOI:** 10.3390/cells13030273

**Published:** 2024-02-01

**Authors:** Maria Ignatyeva, Abdul Kareem Mohideen Patel, Abdulkhaleg Ibrahim, Raed S. Albiheyri, Ali T. Zari, Ahmed Bahieldin, Christian Bronner, Jamal S. M. Sabir, Ali Hamiche

**Affiliations:** 1Département de Génomique Fonctionnelle et Cancer, Institut de Génétique et Biologie Moléculaire et Cellulaire (IG-BMC), CNRS UMR7104, INSERM U964, Université de Strasbourg, 67404 Illkirch, Franceibrahim@igbmc.fr (A.I.); bronnerc@igbmc.fr (C.B.); 2Centre of Excellence in Bionanoscience Research, King Abdulaziz University, Jeddah 21589, Saudi Arabia; ralbiheyri@kau.edu.sa (R.S.A.); azari@kau.edu.sa (A.T.Z.); abmahmed@kau.edu.sa (A.B.); 3Department of Biological Sciences, Faculty of Science, King Abdulaziz University, Jeddah 21589, Saudi Arabia

**Keywords:** HIRIP3, H2A, histone chaperone, CK2 kinase

## Abstract

HIRIP3 is a mammalian protein homologous to the yeast H2A.Z deposition chaperone Chz1. However, the structural basis underlying Chz’s binding preference for H2A.Z over H2A, as well as the mechanism through which Chz1 modulates histone deposition or replacement, remains enigmatic. In this study, we aimed to characterize the function of HIRIP3 and to identify its interacting partners in HeLa cells. Our findings reveal that HIRIP3 is specifically associated in vivo with H2A–H2B dimers and CK2 kinase. While bacterially expressed HIRIP3 exhibited a similar binding affinity towards H2A and H2A.Z, the associated CK2 kinase showed a notable preference for H2A phosphorylation at serine 1. The recombinant HIRIP3 physically interacted with the H2A αC helix through an extended CHZ domain and played a crucial role in depositing the canonical core histones onto naked DNA. Our results demonstrate that mammalian HIRIP3 acts as an H2A histone chaperone, assisting in its selective phosphorylation by Ck2 kinase at serine 1 and facilitating its deposition onto chromatin.

## 1. Introduction

The genome of eukaryotic cells is packaged into chromatin, a highly organized complex of DNA and proteins [1]. Chromatin functions as a scaffold for all the processes of DNA metabolism, such as replication, repair and transcription. Higher-order chromatin structure stability ensures normal cell cycle progression [2], differentiation during development [3] and epigenetic inheritance [4]. There is growing evidence linking deregulated histone synthesis and deposition with human cancers and other diseases [2,5]. There are an extensive number of factors involved in chromatin assembly, with ATP-dependent assembly factors and histone chaperone groups being the most prominent [6]. Histone chaperones are a diverse group of proteins that physically bind histones and regulate their deposition onto chromatin. Their structure is varied, with acidic stretches rich in glutamic and aspartic acids being a common motif. These stretches are often found near the chaperones’ C-terminus. Histone chaperones play a critical role in all the steps of the histone cycle. They protect histones from non-specific interactions and modifications and facilitate histone recruitment to chromatin and are involved in nucleosome incorporation, eviction and exchange [7]. In vitro studies have revealed that at a physiological ionic strength, histone chaperones are necessary to shield the positive charge of histones to prevent the formation of disordered insoluble aggregates between histones and DNA [8]. The histones of the H2A group often interact with chaperones through their C-terminal alphaC helix domain, which was demonstrated on the H2A.Z chaperones YL1 [9], Anp32E [10] and Chz1 [11]. Our understanding of chromatin assembly dynamics relies on the identification of its factors and their functions and interactions. Despite the growing knowledge of histone chaperones, the data on human chaperones are still lacking. In this work, we studied HIRIP3, a mammalian homologue of the yeast histone H2A.Z chaperone Chz1, in order to analyze its specificity toward H2A.Z and determine its biological functions. Indeed, the structural basis for the binding preference of Chz1 for H2A.Z over H2A, as well as the mechanism through which Chz1 modulates histone deposition or replacement, remains enigmatic [12].

HIRIP3 (HIRA-interacting protein 3) is a 556-amino-acid mammalian protein. Its gene is located on the short arm of chromosome 16 and consists of seven exons. The HIRIP3 protein is highly charged, with 20.1% and 21.1% acidic and basic residues, respectively. HIRIP3 mRNAs of fetal and adult origins consist of two low-level transcripts of approximately 2.0 and 2.9 kb [13]. Both transcripts are similarly expressed in fetal kidney, with a predominance of the larger transcript in adult skeletal muscle and the smaller one in adult heart. HIRIP3 is a mainly nuclear protein and was shown to be extensively scattered throughout the nucleus, with nucleoli devoid in interphase cells [14]. During mitosis, HIRIP3 is excluded from condensed chromatin, but is concentrated in mitotic spindles at metaphase and in midbody junctions at the end of telophase [14].

The HIRIP3 domain structure features glutamic-acid-rich and polyserine regions in its N-terminal and central position, respectively, and a CHZ motif located in its C-terminus. The CHZ motif is conserved from yeast to humans and is shown to mediate histone chaperone Chz1 interaction with H2A.Z [15]. Chz1 was reported to prefer H2A.Z–H2B over canonical H2A–H2B [12,15,16]. The basis for such specificity, however, is unclear. The NMR structure of the Chz1 histone-binding region shows how Chz1 recognizes H2A.Z–H2B, but does not clearly reveal how it discriminates H2A.Z from H2A. The CHZ motif in HIRIP3 may also be involved in histone processing. While the roles of the other regions within HIRIP3 are unknown, based on the available data, the serine region may act as the phosphorylation site of HIRIP3 [14], and the glutamic-acid-rich region may mediate its DNA-related functions [17].

HIRIP3 has also been shown to be a HIRA-interacting partner in yeast protein interaction trap studies [13]. HIRA is an H3.3 variant histone chaperone which incorporates H3.3 in replication independently during active transcription [18] and chromatin structure recovery after DNA damage [19]. The HIRA–HIRIP3 interaction was confirmed by in vitro pull-down experiments [13]. HIRA-interacting partners—UBN1, CABIN1 and Asf1a—have been shown to be involved in the chromatin processing, by the means of enhancing and modulating HIRA activity [20,21]. While the HIRA–HIRIP3 complex function is unknown, it is possible that it is associated with chromatin metabolism, similar to other HIRA-containing complexes.

Taking these data together, HIRIP3 is highly implicated in the role of participating in chromatin dynamics. First, it directly interacts with the H3.3-specific chaperone HIRA, the interacting partners of which are often involved in chromatin-associated processes. Second, HIRIP3 interacts with and becomes phosphorylated by CK2 kinase [14], the downstream targets of which have roles in chromatin processing. Finally, HIRIP3 is homologous to the Chz1 yeast protein, which is a histone chaperone specific to H2A.Z.

Despite the available data on HIRIP3, its function has not yet been discovered. In this work, we aimed to identify HIRIP3’s role in the context of chromatin dynamics and to dissect its chaperoning activity. We have found that HIRIP3 is stably associated in vivo with the canonical histone H2A and the CK2 kinase. The recombinant CK2 kinase showed a notable preference for H2A phosphorylation at serine 1. The recombinant HIRIP3 physically interacted with the H2A αC helix through an extended CHZ domain and played a crucial role in depositing canonical core histones onto naked DNA. Taken together, our data establish HIRIP3 as a novel histone chaperone specific to canonical H2A, assisting in its selective phosphorylation at serine 1 and facilitating its deposition onto chromatin.

## 2. Materials and Methods

### 2.1. Immunofluorescence

Immunofluorescence was performed using standard procedures on a Leica DMR microscope (Leica, Wetzlar, Germany) using a 63 3 1.32 NA oil immersion objective.

### 2.2. Protein Expression in Bacteria

Human HIRIP3, H2A, H2A.Z and H2B cDNAs were PCR-amplified and sub-cloned either in a home-made bi-cistronic pET 28b or pGEX-5X1 vectors, which are under the control of T7 promoter [22]. PCR-directed mutagenesis was used to generate all the described mutations or deletions. Constructs were introduced into competent BL21-CodonPlus-RIL bacteria (Stratagene, San Diego, CA, USA) using standard heat-shock transformation procedure [23]. Transformed cells were grown in Luria–Bertani broth at 37 °C, with appropriate antibiotics, till an absorbance of 0.5 at 600 nm. Expression was induced by adding a final concentration of 1 mM isopropyl-β-d-thiogalactopyranoside (Euromedex, Souffelweyersheim, France), which activates expression of genes under control of T7 promoter. After induction, cells were further grown for 2 h at 25 °C. Cell pellets were collected by low-speed centrifugation (10 min at 5000 rpm) and stored at 80 °C.

### 2.3. Protein Expression in HeLa Cells

The coding sequence of HIRIP3 was PCR-amplified and sub-cloned into the XhoI-NotI sites of the pREV-HTF retroviral vector in frame with an N-terminal FLAG and HA tags. Retroviruses were produced in Phoenix retrovirus packaging cells and used to infect target HeLa cells, as described previously [24]. Briefly, Phoenix cells were transfected using pREV-HTF and viral particles were used to transduce HeLa cells. Stably transduced HeLa cell populations were sorted by triple immunomagnetic selection using Dynabeads cd25 (Invitrogen, Waltham, MA, USA). Positive cells were checked by immunofluorescence using standard procedures [25]. Rat anti-HA antibody (Roche, Basel, Switzerland) was used at 1/200 dilution; the secondary antibody goat anti-Rat IgG coupled to Alexa Fluor 488 (Molecular Probes, Eugene, OR, USA) was used at 1/400 dilution. Selected cell line expressing HIRIP3 protein fused to N-terminal FLAG- and HA-epitope tags (e-HIRIP3) was grown in suspension culture to generate four liters. Cells were maintained in incubator equipped with HEPA (high-efficiency particulate arrestance) filter, at 37 °C and 5% CO_2_. Cells were grown in Dulbecco’s Modified Eagle Medium (Thermo Fisher Scientific, Waltham, MA, USA), supplemented with 10% FBS and 0.1 mg/mL penicillin/streptomycin solution. Cell pellets were collected by low-speed centrifugation and stored at −80 °C.

### 2.4. Tandem Affinity Purification from HeLa Cells

Collected HeLa cells pellets were used to prepare three types of soluble extracts: soluble cytosolic extract, soluble nuclear extract and MNase-solubilized chromatin extract, as previously described [26]. e-HIRIP3 complexes were purified by tandem immunoaffinity as previously described [26]. Briefly, extracts were mixed with the appropriate agarose anti-FLAG M2 resin (Sigma, St. Louis, MO, USA) and incubated at 4 °C for at least 4 h. After binding, the extract–resin solution was centrifuged, the supernatant was removed, and the resin was extensively washed with TGEN-150 buffer (20 mM Tris-HCl pH 7.65, 150 mM NaCl, 3 mM MgCl_2_, 0.1 mM EDTA, 10% glycerol, 0.01% NP40, 1 mM PMSF, 1× Roche cOmplete Protease Inhibitor Cocktail). Subsequently, the resin-bound proteins were eluted from the resin by adding FLAG peptide and incubating at 4 °C. The eluates were collected, and if applicable, they were subjected to dialysis against TGEN-150 buffer. A second step of purification was then performed using anti-HA agarose resin (Sigma).

### 2.5. Protein Purification from Bacteria

*E. coli* pellets were resuspended in buffer TGEN-300 (20 mM Tris-HCl pH 7.65, 300 mM NaCl, 3 mM MgCl_2_, 0.1 mM EDTA, 10% glycerol, 0.01% NP40, 1 mM PMSF, 1× Roche cOmplete Protease Inhibitor Cocktail). The lysing process involved freeze–defreeze cycles, followed by two rounds of 2 min sonication at 10 A. The soluble fraction was separated through high-speed centrifugation (12,000 rpm). The soluble proteins were purified on glutathione Sepharose 4B beads (Amersham) or Ni-NTA agarose beads (Qiagen) with standard methods. Briefly, GST- or histidine-fusion proteins were incubated on either glutathione Sepharose or Ni-NTA agarose beads for 4 h at 4 °C in TGN-300 buffer (20 mM Tris, pH 7.65, 3 mM MgCl2, 0.1 mM EDTA, 10% glycerol, and 0.01% NP-40) containing 300 mM NaCl. Beads were then washed extensively in TGN-300. Bound proteins were eluted either with 10 mM glutathione or with 250 mM imidazole. Samples were fractionated on SDS-PAGE and stained with colloidal blue.

Human recombinant wild-type or mutant histone dimers FLAG-H2A/HIS-H2B or FLAG-H2A.Z/HIS-H2B were co-expressed in BL21-CodonPlus-RIL (Stratagene), Nickel agarose purified and stored in 50 mM Tris-HCl pH 7.65, 1 M NaCl; 0.1 mM EDTA; 1 mM PMSF.

### 2.6. SDS-Polyacrylamide Gel Electrophoresis (SDS-PAGE)

Protein eluates were mixed with SDS-PAGE loading buffer, heated for 5 min at 95 °C and loaded onto the 12% Tris-Glycine or 4–12% Bis-Tris polyacrylamide gel. Proteins were separated at 180 V until the dye front had reached the bottom of the gel. The molecular weight of proteins was estimated by running unstained (Mark12 Unstained Standard, Thermo Fisher Scientific) and pre-stained (PageRuler Plus, Thermo Fisher Scientific) marker proteins. Following electrophoresis, proteins were stained with either Coomassie Brilliant Blue, silver or subjected to Western blotting.

### 2.7. Staining of Protein Gels

For Coomassie staining, polyacrylamide gels were fixed for at least 30 min in fixation solution (30% ethanol/10% acetic acid). Afterwards, they were incubated in Coomassie staining solution (0.025% Coomassie Blue R-250 in 10% acetic acid) for 20 min on a slowly rocking platform. To enhance visualization of proteins, gels were destained by boiling in MilliQ water and further incubation on the rocking platform.

The staining of protein gels with silver nitrate solution was carried out using SilverQuest Kit (Thermo Fisher Scientific).

### 2.8. Western Blotting

Western blotting was carried out using standard procedures [27]. Proteins were separated by SDS-PAGE and transferred to nitrocellulose or PVDF membranes using the Bio-Rad blotting system. Antibodies used in this work were as follows: 3F10, monoclonal anti-HA 000000011867423001, Roche; A3S monoclonal anti-H3 CT pan 05–928, Upstate Biotechnologies; polyclonal anti-H2B 07–371, Upstate Biotechnologies; polyclonal anti-H2A.Z, kind gift by Stefan Dimitrov; polyclonal anti-H2A, kind gift by Stefan Dimitrov; polyclonal anti-CK2 alpha A300-197A-M, Bethyl Laboratories; monoclonal M2 anti-FLAG A8592, Sigma Aldrich.

### 2.9. In-Gel Histone Deposition Assay

Increasing amounts of recombinant HIRIP3 mixed with stoichiometric amounts of core histones were incubated with 2 μg of 147 bp 601 DNA for 30 min at 37 °C in 20 mM Tris-HCl (pH 7.5), 50 mM NaCl, 3 mM MgCl_2_ and 0.01% BSA. The reaction products were analyzed on 5% native polyacrylamide gel 1 × TG (0.025 M Tris-HCl, 0.192 M glycine), run at room temperature for 55 min at 120 V. Control nucleosomes were assembled on 147 bp 601 DNA using to the “salt jump” method [28] and a histone-to-DNA ratio rw = 1.

### 2.10. Mass Spectrometry

Identification of proteins was carried out using an Orbitrap mass spectrometer (ThermoFinnigan) by Taplin Biological Mass Spectrometry Facility (Harvard Medical School, Boston, MA, USA) [10].

### 2.11. Structural Modelling

Chz1 and H2A.Z/H2B fusion structures were used from 2JSS [16], the structure of budding yeast chaperone Chz1 complexed with fusion histone H2A.Z-H2B. H2A/H2B dimer structure was used from 1ID3 [29], the yeast nucleosome particle. Figures have been made using PyMOL program (The PyMOL Molecular Graphics System, Version 1.3 Schrödinger, LLC.), http://www.pymol.org/ (accessed on 15 March 2023). The structures were superimposed using alignment of H2B region identical in both structures.

### 2.12. Kinase Assay

Phosphorylation assay of H2A/H2B, S1A H2A/H2B and S122A H2A/H2B was carried out as described previously [14] with minor modifications. A total of 1 μg of each of the protein complexes was incubated for 30 min at 30 °C in a buffer containing 70 mM Tris-HCl, 100 mM NaCl, 10 mM MgCl_2_, 5 mM DTT, 0.5 m Ci P32 ATP and 100 ng of each of recombinant CK2α1/β and CK2α2/β in the absence or presence of 10 μM silmitasertib (CX-4945). Recombinant CK2α1/β (CSNK2A1/B) 05–184 and CK2α2/β (CSNK2A2/B) 05–185 were purchased from Carna Biosciences. Reactions were stopped by Laemmli sample buffer, boiled for 10 min, migrated through a 1% SDS-polyacrylamide gel, dried on paper and revealed by exposure to radiography film.

## 3. Results

### 3.1. H2A/H2B Dimers Co-Purify with HIRIP3 In Vivo

The goal of this work is to investigate the function of HIRIP3 and to evaluate its chaperoning specificity toward H2A.Z. We aimed to test whether HIRIP3 is directly associated with histones, since physical interaction with histones is an important indicator of histone chaperone activity [7]. Towards this goal, we established a HeLa cell line stably expressing N-terminally FLAG-HA epitope-tagged human HIRIP3 (e-HIRIP3). The cell line was verified to be positive for HIRIP3 expression by immunofluorescence with an anti-HA antibody (Figure 1A). We next prepared cytosolic, nuclear soluble and chromatin extracts from the e-HIRIP3 cell line and subjected them to tandem anti-FLAG and anti-HA affinity purification, which allowed us to identify proteins directly interacting with e-HIRIP3 (Figure 1B). In the cytosol, e-HIRIP3 was found to form a cytosolic complex (CC) with the casein kinase CK2 alpha (CSNK2A1 or CSNK2A2) and beta subunits (Figure 1B), as validated by mass spectrometry (Figure 1C) and Western blotting analysis (Figure 1D). In the nucleus, e-HIRIP3 was found to form a soluble nuclear complex (SNC) and a chromatin-associated complex (CAC) containing in addition to the CK2 kinase, the four nucleosomal core histones (H2A, H2B, H3 and H4), the FACT complex (Spt16 and SSRP1) and the DNA-dependent protein kinase (PRKDC) complex with its associated cofactors, Ku70/80 and PARP1, (Figure 1B). The mass spectrometry and Western blotting analysis of the purified SNE and CAC e-HIRIP3 complexes confirmed their association with the canonical histones H2A–H2B and Ck2 kinase (Figure 1C,D). Surprisingly, H2A.Z was not found to be associated with the complex neither by mass spectrometry nor by Western blotting (Figure 1C,D). Our data show that, unlike yeast Chz1, which interacts with H2A.Z, HIRIP3 associates exclusively with the canonical H2A in vivo. The e-HIRIP3 complex was also found to associate with the FACT complex, which is known to bind to the H3 and H4 N-terminal tails, thus explaining their presence in the soluble nuclear complex (Figure 1B–D). These findings reveal that HIRIP3 is stably associated with histones in vivo and may have a function related to their metabolism.

### 3.2. HIRIP3 Physically Interacts with H2A/H2B and H2A.Z/H2B Dimers In Vitro through Its CHZ Motif

The domain structure of HIRIP3 includes glutamic-acid-rich and polyserine regions located in its N-terminal and central positions, respectively. Additionally, there is a CHZ motif at its C-terminus (Figure 2A). This CHZ motif is conserved from yeast to humans and plays a crucial role in facilitating the interaction between the yeast histone chaperone Chz1 and H2A.Z [12,15,16].

To further investigate the interaction between HIRIP3 and histones, we used an in vivo *E. coli* co-expression system based on a homemade pET bicistronic vector [9,30]. In this setup, GST-HIRIP3 was co-expressed in *E. coli* BL21 with either Flag-H2A/His-H2B or Flag-H2A.Z/His-H2B dimers. The pull-down experiments using GST-HIRIP3 followed by Flag-H2A/H2A.Z demonstrated that HIRIP3 binds to both the H2A/H2B and H2A.Z/H2B dimers with a similar affinity irrespective of the salt concentration used (Figure 2B). This result suggests that HIRIP3 has the potential to interact with both H2A and H2A.Z, similar to the histone chaperone Nap1 [31].

However, when tested in HeLa cells, H2A.Z was not detected within the e-HIRIP3 complex. This indicates that HIRIP3 either does not interact with H2A.Z in vivo or the interaction is too transient to be detected. Overall, our data indicate that HIRIP3 stably interacts with H2A/H2B dimers in vivo, despite its ability to bind both H2A/H2B and H2A.Z/H2B dimers in vitro.

To further investigate the interaction between HIRIP3 and H2A/H2B, we aimed to map the specific HIRIP3 domain responsible for histone recognition and binding. To achieve this goal, we conducted a structure-based mutational analysis of HIRIP3. Initially, we generated two mutants, each encompassing approximately half of the protein. The first mutant consisted of residues 1–370, which included the N-terminal regions unique to HIRIP3. Subsequently, we generated the second mutant containing residues 371–556 of HIRIP3 (including the CHZ motif at positions 484–507) and its surrounding regions (Figure 2C). Next, we co-expressed these mutants with H2A/H2B dimers and prepared soluble bacterial extracts for analysis. Through a GST pull-down assay, we discovered that the 371–556 mutant displayed an interaction with the H2A–H2B dimers with an affinity similar to that of the full-length HIRIP3 protein, while the 1–370 mutant exhibited no interaction (Figure 2C).

To determine the minimal HIRIP3 domain required for interaction with H2A, we further truncated the region spanning residues 371–556 from both the N and C-terminal ends, as illustrated in Figure 2A. The N-terminally truncated mutant, spanning residues 403–527, displayed a complete affinity for interacting with the H2A/H2B dimers, while the interaction of the mutant encompassing residues 425–527 showed a significant decrease (Figure 2D, comparing lanes 3 and 4). Similarly, the C-terminally truncated mutant covering residues 403–514 exhibited a reduced affinity (Figure 2D, lane 6). These experiments enabled us to identify the region 403–527 of HIRIP3 as the minimal domain sufficient for a full-affinity interaction with the H2A/H2B dimers. We will henceforth refer to this minimal H2A/H2B binding region (403–527) as the extended CHZ motif.

The next step was to assess whether the CHZ motif is indispensable for HIRIP3 interaction with H2A/H2B, or if this interaction could be attributed solely to regions outside the CHZ motif. Towards this goal, we constructed a deletion mutant of HIRIP3, lacking only the CHZ domain (484–507), which was previously identified as essential for interaction with yeast H2A.Z/H2B. The resulting mutant, HIRIP3_ΔCHZ, failed to interact with the H2A–H2B dimers, in contrast to the strong interaction observed with the wild-type protein in the same GST pull-down assay (Figure 2E). This finding confirms that the CHZ motif of HIRIP3 is indispensable for its interaction with H2A–H2B.

Finally, we examined whether the CHZ domain is necessary for HIRIP3’s histone deposition activity. For this purpose, we compared the ability of wild-type HIRIP3 and a mutant lacking the CHZ domain to deposit recombinant canonical core histones onto DNA using an electrophoretic mobility shift assay (EMSA) (Figure 2F). Under these conditions, HIRIP3 lacking the CHZ domain exhibited a diminished deposition activity (Figure 2F, lanes 3–5) compared to the full-length protein (Figure 2F, lanes 6–7). Based on these findings, we concluded that the CHZ domain is required for HIRIP3s chaperoning activity.

### 3.3. H2A Interacts with HIRIP3 through Its alphaC Domain

After mapping the minimal interaction region of HIRIP3, we proceeded to identify the specific domain of H2A involved in this interaction. To achieve this, we generated a series of H2A truncation mutants affecting the C-terminus of the protein: 1–123, 1–111, 1–98 and 1–92 (Figure 3A). The choice of truncations was based on the existing literature data that highlight the importance of the H2A alphaC helix for interactions with various histone chaperones [9,10,11].

Next, we co-expressed the GST-H2A truncation mutants along with the FLAG-HIRIP3 (371–556) region, which contains the CHZ minimal domain and the surrounding regions, and performed anti-FLAG pull-down assays to assess their interaction. The results indicated that residues 92–111 of H2A were crucial for the interaction with HIRIP3 (Figure 3B). Notably, this region contains the alphaC helix, which has been previously shown to be responsible for interactions with several H2A-group chaperones, such as Chz1 [11], YL1 [9] and Anp32E [10]. This finding suggests that HIRIP3 interacts with H2A through its alphaC helix, characteristic of H2A-group histone chaperones.

### 3.4. Structural Considerations of HIRIP3 and H2A/H2B Interaction

Our next step was to investigate the structural factors influencing the interaction between the CHZ motif and histones. To visualize the interactions of HIRIP3 with both H2A/H2B and H2A.Z/H2B, we utilized the available structural data [16]. We superimposed the yeast Chz1 and H2A.Z–H2B fusion structure (2JSS) [16] onto the H2A/H2B dimers from the human nucleosome particle structure (6KE9) [32] (Figure 4A). The goal was to determine whether the CHZ motif’s interaction with histones is dependent on structural differences between H2A/H2B and H2A.Z/H2B.

We found that the alignment of the CHZ motif with H2A/H2B was very close to that with H2A.Z–H2B. The CHZ motif lies on top of the H2B alphaC helix and bends around it, reaching the H2A alpha2 helix. Interestingly, the alphaC helix of H2A does not directly participate in the interaction with the CHZ motif. This indicates that HIRIP3 utilizes other residues from the 403–527 minimal domain for this interaction.

Upon analyzing the residues involved in the interaction, we observed that the CHZ motif makes direct contacts with the conserved regions of both yeast H2A and H2A.Z, specifically Ser 47 and Glu 66 (Figure 4B,C). However, Ser 47 is mutated to Alanine in both human H2A and H2A.Z (Figure 4B,D). On the basis of these data, while the CHZ motif constitutes a core region for interactions with histones, it does not appear to define the specificity of the interaction between the H2A/H2B and H2A.Z/H2B dimers.

Instead, we propose that HIRIP3’s specific in vivo interaction with H2A, rather than H2A.Z, likely originates from the surrounding CHZ regions. Other non-structural determinants, such as complex co-factors or post-translational modifications (PTMs), could also play a role in conferring the observed H2A specificity in vivo. These supplementary factors may modulate HIRIP3’s selectivity in interacting with H2A, thereby contributing to its distinct functional roles in cellular processes.

### 3.5. Recombinant CK2 Phosphorylates H2A at Serine 1

The mass spectrometry analysis of the cytosolic HIRIP3 complex (CC) revealed the presence of the CK2 kinase within the complex (Figure 1C). The Western blotting against the alpha subunit of CK2 confirmed CK2 as being part of the HIRIP3 complex (Figure 1D). The presence of CK2 in the HIRIP3 complex could either play a role in HIRIP3 phosphorylation or have other unknown functions in the context of the HIRIP3 complex. To test the latter possibility, we used an in-gel phosphorylation assay to evaluate whether CK2 can phosphorylate the H2A/H2B dimers. Toward this goal, we generated two H2A non-phosphorylable mutants (S1A and S122A) (Figure 5A). The phosphorylation assay revealed that CK2 can phosphorylate the full-length H2A histone and S122A mutant but not the S1A mutant (Figure 5B). The S1A mutation completely abolished the phosphorylation of H2A, indicating that the Serine 1 site is critical for H2A phosphorylation, even though it is not a canonical site for the Ck2 kinase. The most crucial specificity determinant is an acidic residue at position +3 but additional acidic residues at positions spanning from −2 to +7 (and probably farther) could also act as positive specificity determinants for CK2 [33]. In addition, Ck2 has been shown to phosphorylate noncanonical sequences [34]. Accordingly, the incubation of Ck2 with silmitasertib (CX-4945), a CK2-specific inhibitor, completely abolishes H2A phosphorylation (Figure 5C). The presence of HIRIP3, CK2 and H2A/H2B in one stable complex implicates a functional connection between those factors that may be regulated through phosphorylation.

## 4. Discussion

In this study, we explored the molecular function of human HIRIP3 and identified its interacting partners. The process of purifying e-HIRIP3 from various compartments within HeLa cells yielded the discovery of numerous interacting partners. This finding provides a suggestive indication of the intricate biological functions associated with HIRIP3.

### 4.1. HIRIP3 Interaction with H2A/H2B Dimers In Vivo and In Vitro

The nuclear soluble and chromatin-associated HIRIP3 contained H2A and H2B histones, but not H2A.Z. Given that H2A and H2B have a strong affinity for each other and function as dimers [35], we propose that HIRIP3 associates with them as a H2A/H2B dimer complex. The in vivo association of HIRIP3 with H2A/H2B supports the idea of their functional connection, as complex formation often indicates functional cooperation. For instance, other chaperones involved with H2A-group/H2B histones, like YL1 [9] and ANP32e [10], have been found to co-purify with the histone variant H2A.Z. Additionally, along with H2A and H2B, H3/H3 histones were also identified within the soluble e-HIRIP3 complex. This presence might suggest HIRIP3’s interaction with the full octamer. It is possible that the Spt16/SSRP1 complex, which has a strong affinity for H3, contributes to this association. The presence of the DNA-dependent protein kinase catalytic subunit (PRKDC) and its associated cofactors, Ku70/80 and PARP1, might stabilize core histones within the complex, suggesting a potential involvement of HIRIP3 in the non-homologous end joining (NHEJ) DNA repair pathway. The identified generic chaperone NPM1 could also aid in stabilizing the complex and contributing to histone deposition. The presence of AMP deaminase (AMPD2) in the complex is intriguing. AMPD2 catalyzes the deamination of AMP to IMP and plays a role in the purine nucleotide cycle. This association hints at a possible critical role of HIRIP3 in energy metabolism.

Furthermore, through in vitro purification assays, we confirmed HIRIP3’s interaction with H2A/H2B dimers. HIRIP3 did not distinguish between H2A/H2B and H2A.Z/H2B dimers in vitro, binding to them with similar affinity. The observation that HIRIP3 associates with H2A/H2B in vivo, unlike yeast Chz1, which associates with H2A.Z/H2B dimers, raises questions about evolutionary divergence in CHZ motif-containing proteins. Notably, neither human HIRIP3 nor Arabidopsis thaliana Chz1 contains the H2A.Z G98/A57-binding domains and DEF/Y motif critical for H2A.Z/H2B dimer binding [12]. This divergence could be attributed to distinct cellular requirements between yeast and mammals. The regions in HIRIP3 absent in Chz1 might mediate this specificity. These regions could serve as scaffolds for interactions with other proteins, such as HIRA, which interacts with HIRIP3 [13]. They might also harbor motifs susceptible to post-translational modifications, contributing to unique HIRIP3 interactions and functions. Accordingly, HIRIP3 is phosphorylated by CK2 kinase [14]. CK2 phosphorylation could influence HIRIP3’s histone specificity, given its role in regulating histone chaperone localization and function, as shown for the H2A histone chaperones Nap1 [36] and FACT [37]. Notably, while Chz1 is known as an H2A.Z chaperone in vivo, it can also interact with H2A in vitro [15]. This supports the idea that both Chz1 and HIRIP3, in vitro, lack the ability to discriminate among H2A-group histones. In vivo, these interactions are likely modulated by other cellular factors, leading to the observed histone specificity.

The chromatin-associated HIRIP3 complex (CAC) contained the four canonical core histones in addition to the other components described above. The association of HIRIP3 with the nucleosome could be mediated by the acidic pocket formed by the H2A–H2B acidic patch. Many chromatin proteins use a conserved arginine residue, known as the arginine anchor, to associate with the acidic patch. Accordingly, yeast Chz1 was found to bind the H2A.Z(1–114) truncation but not the H2A.Z(1–104) truncation lacking the alphaC [11]. In summary, our data establish HIRIP3 as a novel interaction partner of the H2A/H2B canonical histone dimer both in vivo and in vitro.

### 4.2. Structural Basis of HIRIP3 Interaction with H2A/H2B Dimer

This work has demonstrated that mammalian HIRIP3 interacts with H2A in a manner similar to how yeast Chz1 interacts with H2A.Z, facilitated by its conserved CHZ motif. The structural simulations support the similarities between these two interactions, as the CHZ motif closely aligns with both H2A.Z/H2B and H2A/H2B dimers. In contrast to the Chz1–H2A.Z interaction, which involves a 55-residue region of Chz1 [15], we have shown that HIRIP3 engages with H2A through a more extensive 124-residue region. This 403–527 region contains additional residues both on the N- and C-terminal sides of the CHZ motif. These regions likely contribute to a larger interaction interface between HIRIP3 and H2A, involving additional connections beyond the CHZ motif. Accordingly, OsChz1 binds both the H2A–H2B and H2A.Z–H2B dimers in vitro, exhibiting a higher affinity for the canonical H2A than for the variant H2A.Z. An extended Chz1 domain, spanning amino acids 338–471, was shown to have a higher binding affinity for H2A–H2B compared to H2A.Z–H2B [38].

Despite the involvement of other regions, our findings underscore the indispensability of the CHZ motif for HIRIP3’s interaction with H2A/H2B. We propose that the CHZ-motif-containing domain of HIRIP3 employs its entire surface to engage with H2A/H2B dimers.

The structural simulations of the CHZ motif and its interaction with H2A/H2B suggest the direct involvement of both the H2A and H2B histones in binding to HIRIP3. The alphaC region of H2B provides an extensive binding surface for the CHZ motif, while alpha2 of H2A establishes multiple contacts with it. The H2A group of histones is highly divergent and often dictates specificity to chaperones [39], as observed with the YL1 chaperone [9]. While H2B histones might not contribute to specific recognition by chaperones in the same manner as H2A-group chaperones, they may function as scaffolds for chaperone-histone binding or serve other roles. Notably, the highly basic domain of the H2B N-terminal tail plays a vital role in nucleosome assembly by the FACT histone chaperone, possibly by stabilizing nascent nucleosomes [40]. The CHZ motif also exhibits a specific charge distribution that facilitates its interaction with residues of both H2A.Z and H2B histones [16]. As a homologue of Chz1, HIRIP3 likely exhibits similar interaction dynamics with H2A and H2B, employing the charged CHZ motif and other regions within 403–527 to extensively interact with both histones.

Finally, our research demonstrates that HIRIP3 interacts with the C-terminal alphaC helix region of H2A, a characteristic shared with several other chaperones, including yeast Chz1 [11]. Structural simulations indicate that the CHZ motif does not directly bind to either H2A.Z or the H2A alphaC helix, highlighting the importance of regions beyond the CHZ motif for this interaction. Collectively, our data establish the 403–527 region of HIRIP3 as the minimal domain necessary for interacting with the alphaC region of H2A and other regions within the H2A/H2B dimer.

### 4.3. CK2 Kinase Function within HIRIP3 Complex

In the cytosol, e-HIRIP3 was found to be exclusively associated with the cyclin-dependent kinase complex (CK2). CK2 is an evolutionarily conserved ubiquitous serine/threonine kinase that forms a tetrameric complex consisting of two catalytic subunits (CSNK2A1/CSNK2A2) and two regulatory subunits (CSNK2B) [41]. CK2 is involved in a complex series of cellular events, including gene regulation, cell cycle progression, and the maintenance of cell viability. This is achieved by facilitating the protection of cellular proteins from caspase activity. Previous research has demonstrated the interaction between HIRIP3 and CK2 kinase, as well as the phosphorylation of HIRIP3 by CK2 kinase [14]. Two-hybrid interaction studies have revealed both CK2 alpha and beta subunits as partners interacting with HIRIP3 [14]. The HIRIP3 protein undergoes significant phosphorylation in vivo and is influenced by phosphatase treatment, resulting in faster migration on SDS-PAGE gel [14]. The minimal consensus site for CK2-mediated phosphorylation is present multiple times in HIRIP3. Consequently, HIRIP3 has been shown to undergo phosphorylation by CK2 both in vitro and in vivo [14], establishing itself as a substrate for CK2 kinase.

This work suggests a potential functional connection between HIRIP3 and H2A phosphorylation. We demonstrated that CK2 kinase phosphorylates H2A at the S1 position, despite it not being a canonical site for CK2 kinase. This result is not surprising given that CK2 has been reported to phosphorylate noncanonical sequences [34]. The incubation of CK2 with silmitasertib (CX-4945), a CK2-specific inhibitor, completely abolishes H2A phosphorylation (Figure 5C), thereby validating this specificity for a non-canonical substrate. The presence of CK2 kinase and its phosphorylation of both HIRIP3 and H2A raises the possibility of a close interplay within the HIRIP3 complex, mediated by phosphorylation. Accordingly, H2AS1 phosphorylation has been shown to act as a pre-deposition modification that signals for the proper dimerization and deposition of H2A–H2B [42]. However, we cannot rule out the possibility that HIRIP3 is the main target of CK2 [14]. Post-translational modifications are known to function as molecular switches for histone activities. For instance, the phosphorylation of H2A.X is essential for its functionality during DNA repair [43], and the acetylation/deacetylation processing of H3/H4 tetramers is indispensable for their incorporation during de novo chromatin assembly [44]. Based on the data showing HIRIP3’s interaction with H2A/H2B dimers (from this study) and the fact that CK2 was found to associate with H2A [45], we propose that HIRIP3, H2A/H2B and CK2 may constitute a cross-interacting cooperative complex.

## 5. Conclusions

Based on our data, we propose HIRIP3 as a histone chaperone that specifically interacts with H2A histones. Firstly, it associates with H2A/H2B dimers both in vitro and in vivo. Secondly, HIRIP3 interacts with histones through an extended CHZ domain, encompassing both the CHZ motif and its surrounding region. Thirdly, the interaction between HIRIP3 and H2A relies predominantly on the alphaC helix, which is a distinctive feature of histone chaperone interactions with H2A-group histones. Finally, HIRIP3 facilitates the recruitment of CK2 kinase to phosphorylate Serine 1 of H2A.

In summary, our study establishes the mammalian HIRIP3 protein as a novel histone chaperone specific to H2A. It achieves this specificity by utilizing an extended CHZ domain spanning the HIRIP3 403–527 region, which enables interaction with the C-terminal alphaC helix of H2A.

## Figures and Tables

**Figure 1 cells-13-00273-f001:**
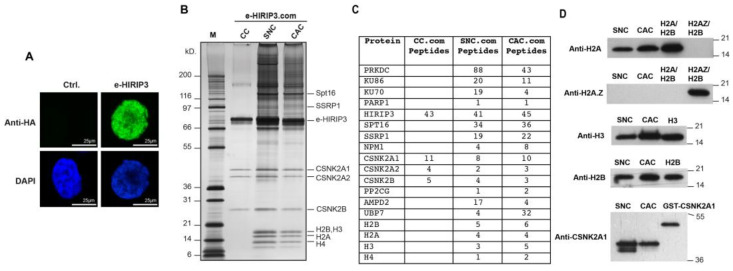
H2A/H2B dimers co-purify with HIRIP3 in vivo. (**A**) Stable expression of e-HIRIP3 in HeLa cells. Cells expressing e-HIRIP3 and control cells (Ctrl) were stained with anti-HA (**top**) and DAPI (**bottom**). (**B**) Silver staining of e-HIRIP3 complexes purified from HeLa cells stably expressing e-HIRIP3, separated by fractionation (CC—cytosolic complex, SNC—soluble nuclear complex, CAC—chromatin associated complex). (**C**) Mass spectrometry table of the different e-HIRIP3 complexes purified from HeLa cells. (**D**) Western blotting analysis of the SNC and CAC e-HIRIP3 complexes. Complexes were probed with anti-H2A, anti-H2A.Z, anti-H3, anti-H2B and anti-CK2 antibodies. The last slot in each experiment is an independently purified protein used as an antibody positive control. The control used for CK2 alpha subunit is GST-CK2 alpha fusion protein, which is 26 kDa larger than endogenous CK2 alpha.

**Figure 2 cells-13-00273-f002:**
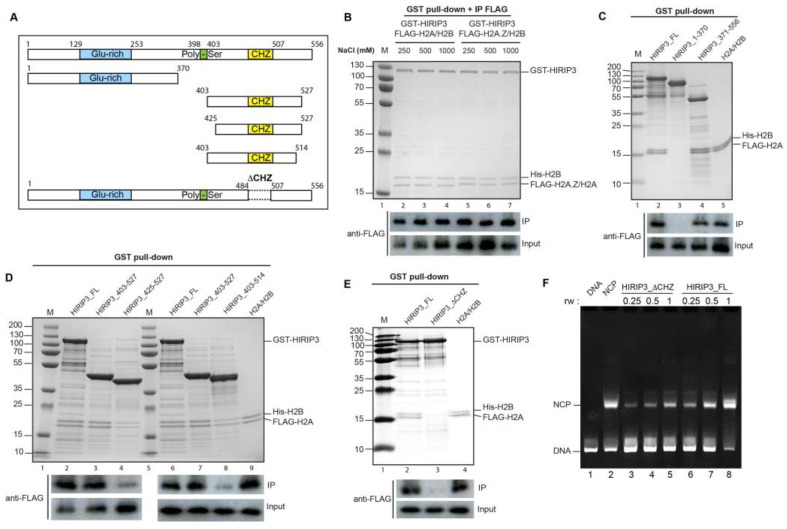
HIRIP3 interacts with both H2A/H2B and H2A.Z/H2B dimers in vitro. (**A**) Schematic representation of full-length HIRIP3 and HIRIP3 mutants used in this work (HIRIP3 1–370, HIRIP3 371–556, HIRIP3 403–527, HIRIP3 403–514, HIRIP3 425–527 and HIRIP3 without the CHZ domain (1–484 fused to 507–556). Glutamic-acid-rich region (blue), polyserine region (green), CHZ motif (orange). (**B**) (**Top**): Coomassie blue stained SDS-PAGE showing pull-down of full-length GST-HIRIP3 co-expressed in bacteria with either FLAG-H2A/HIS-H2B or FLAG-H2A.Z/HIS-H2B dimers. Soluble cellular extracts were isolated and subjected to GST pull-down and FLAG immunopurification. M: MW—molecular weight marker; GST-HIRIP3 (full-length HIRIP3 1–556); Flag-H2A(Z)/His-H2B control (purified FLAG-H2A(Z)/HIS-H2B used as a protein molecular weight marker). (**Bottom**): Western blot against FLAG-H2A using the input or the IP. (**C**) (**Top**): Coomassie blue stained SDS-PAGE showing pull-downs of GST-HIRIP3 FL (full-length), GST-HIRIP3 (1–370) and GST-HIRIP3 (371–556) co-expressed with FLAG-H2A/HIS-H2B dimers. (**Bottom**): Western blot against Flag-H2A using the input or the IP. (**D**) (**Top**): Coomassie blue stained SDS-PAGE showing pull-downs of GST-HIRIP3 (403–527), GST-HIRIP3 (425–527) and GST-HIRIP3 (403–514) co-expressed with FLAG-H2A/HIS-H2B dimers. (**Bottom**): Western blot against FLAG-H2A using the input or the IP. (**E**) (**Top**): Coomassie blue stained SDS-PAGE showing pull-downs of GST-HIRIP3 full-length or mutant lacking the CHZ motif (DCHZ 484–507). (**Bottom**): Western blot against FLAG-H2A. (**F**) Ethidium bromide-stained native PAGE showing the histone deposition activity of GST-HIRIP3 full-length or a mutant lacking the CHZ motif in the presence of an increasing histone/DNA ratio (rw). NCP: Nucleosome Core Particle; DNA: 147bp 601 DNA.

**Figure 3 cells-13-00273-f003:**
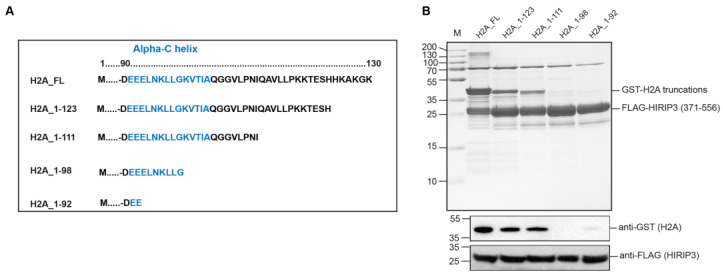
HIRIP3 interacts with H2A through its alphaC helix. (**A**) Schematic representation of full-length H2A and H2A truncations used in this work: H2A 1–130 (FL, full-length), H2A 1–123, H2A 1–111, H2A 1–98 and H2A 1–92. AlphaC helix region is indicated in blue. (**B**) (**Top**): FLAG purification assay of H2A-GST truncations with 371–556 FLAG-HIRIP3. Interaction is gradually decreased from full-length to 1–111 truncation and is lost for 1–98 mutant. M: molecular weight marker. (**Bottom**): Western blot against Gst-H2A and Flag-HIRIP3 using the IP.

**Figure 4 cells-13-00273-f004:**
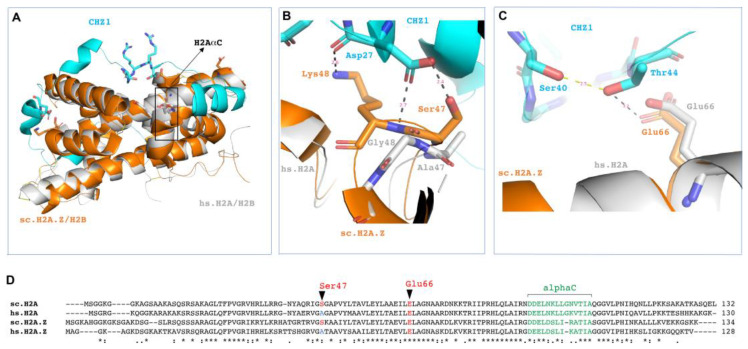
Structural determinants of HIRIP3 interaction with H2A/H2B and H2A.Z/H2B. Superposition of Saccharomyces H2A.Z/H2B (orange) with CHZ1 (Teal) on human H2A/H2B (grey), showing the overall histone fold is similar. H2A aplhaC shown in the highlighted box (deep yellow) (**A**). Ser47 and Lys48 of yeast H2A.Z (orange), key residues interacting with the Asp27 sidechain and the backbone of CHZ1 (Teal), respectively. Human H2A/H2B (grey) has Ala47 and Gly48 at this position (**B**). Superimposition of human histone H2A (grey) on saccharomyces histone H2A.Z (orange) at position Glu66 shows possible interaction with Thr44 sidechain of CHZ1 (**C**). Sequence alignment of Saccharomyces H2A.Z, human H2A.Z and human H2A shows the dissimilar amino acid at position 47 (Ser) and the identical amino acid at position 66 (Glu) (**D**). Multiple amino acid sequence alignments between yeast sc.H2A and sc.H2A.Z and human hs.H2A and hs.H2A.Z. Protein alignment was performed using ClustalW software (latest v. 2.1) (EMBL-EBI). Amino acids interacting with Chz1 are highlighted with red background and the H2A/H2A.Z alphaC with green background.

**Figure 5 cells-13-00273-f005:**
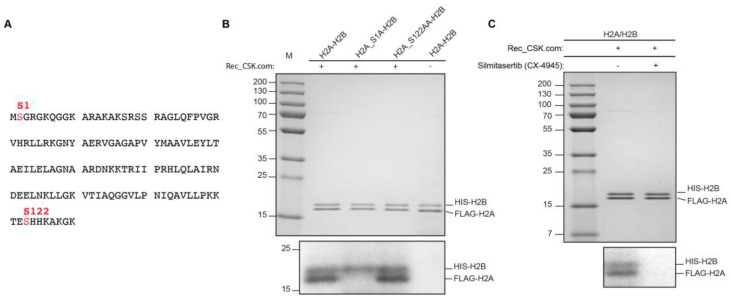
(**A**) Recombinant CK2 phosphorylates H2A at serine 1. Wild-type and mutant (S1A and S122A, highlighted in red) human H2A sequences used in the kinase assay. (**B**) (**Top**): A total of 1 μg of each of H2A/H2B, S1A H2A/H2B and S122A H2A/H2B recombinant protein were incubated at 30 °C for 30 min in the presence of CK2 kinase and [γ-^32^P] ATP. (**Bottom**): Following incubation, the reactions underwent electrophoresis on a 12% SDS-polyacrylamide gel and were subsequently visualized through exposure to radiography film. (**C**) (**Top**): a total of 1 μg of recombinant H2A/H2B dimers were incubated at 30 °C for 30 min in the presence of CK2 kinase and [γ-^32^P] ATP in the absence or presence of 10 μM silmitasertib (CX-4945) inhibitor. (**Bottom**): Following incubation, the reactions underwent electrophoresis on a 12% SDS-polyacrylamide gel and were subsequently visualized through exposure to radiography film.

## Data Availability

All the data supporting the reported results are included in this manuscript.

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
