# Peer review of "Identification and Characterization of HIRIP3 as a Histone H2A Chaperone"

_cells, 2024, doi:10.3390/cells13030273_

Round 1

Reviewer 1 Report

Comments and Suggestions for Authors

 In the manuscript, the authors analyzed function of mammalian HIRIP3 and identified its interacting partners. They found that HIRIP3 is specifically associated in vivo with the H2A-H2B dimers and CK2 kinasefurthermore, HIRIP3 physically interacted with the H2A αC helix through an extended CHZ domain. The 403-527 region of  HIRIP3 is the minimal domain necessary for interacting with the alphaC region of H2A. In the cytosol, HIRIP3   exclusively associated with CK2. CK2 phosphorylated H2A at the S1 position. Finally, the authors proposed that HIRIP3, H2A/H2B, and CK2 may constitute a cross-interacting cooperative complex.

 Minor revision:

1. In Key words,  “H2A” is better than “H2A.Z”.

2. In Line 111, “Cell pellets were collected by low-speed centrifugation....”, centrifugal force should be indicated.

3. In Figure 1A, the scale bar should be indicated.

4. Line 303-307, “Under these conditions, HIRIP3 lacking the CHZ domain exhibited significantly impaired deposition activity (Fig. 2E, lanes 3-5) compared to the full-length protein (Fig. 2F, lanes 6-7).” Fig.2E should be Fig.2F.

5. I n Figure 5., the author should give title and describe “A”content.

Author Response

We thank this review for his/her positive comments. We have followed the reviewer's recommendations and we believe that this has strengthened the manuscript.

 Minor revision: 

  1. In Key words,  “H2A” is better than “H2A.Z”.

A>) This was changed in the new version of the manuscript

  1. In Line 111, “Cell pellets were collected by low-speed centrifugation....”,centrifugal force should be indicated.

A> This was corrected and reads: “Cell pellets were collected by low-speed centrifugation (10 min at 5000 rpm)”.

    3.  In Figure 1A, the scale bar should be indicated.

A> Scale bars are now included.

  1. Line 303-307, “Under these conditions, HIRIP3 lacking the CHZ domain exhibited significantly impaired deposition activity (Fig. 2E, lanes 3-5) compared to the full-length protein (Fig. 2F, lanes 6-7).” Fig.2E should be Fig.2F. 

A>) This was corrected in the new version of the manuscript

  1. In Figure 5., the author should give title and describe “A”content.

A>) This was corrected and a tittle is included.

Reviewer 2 Report

Comments and Suggestions for Authors

HIRIP3 is HIRA interacting protein, previous works have shown that HIRIP3 binds to H2B, and it contains CHZ domain, indicated as a histone chaperone. In this manuscript, Ignatyeva et al did some in vivo and in vitro pulldown experiments to demonstrate that human HIRIP3 is also an H2A chaperone. Overall this manuscript contains data that is clear but loses clear connection between the data and the conclusion they want to draw.

For example, the authors proposed the regions surrounding CHZ domain might be responsible for the specific preference of H2A but did not elaborate further with experiments to demonstrate that. I believe these experiments are crucial to understand the molecular mechanism the authors are trying to establish. The big difference between H2A and H2A.Z comes from this extended C terminus. The authors shall investigate further to demonstrate in a molecular level that such preference is related to the surrounding region. I feel that the authors are so close to the answer but somehow stopped moving forward further. For example, a comparison of H2A and H2A.Z pulldown with different HIRIP3 constructs would help explain why HIRIP3 has a preference on H2A in vivo.

Therefore, I would recommend some revisions based on the comments here, including the minor issues listed below.

Minor error:

Previous works have shown that HIRIP3 binds to H2B, would this explain the minimal difference between H2A-H2B vs H2A.Z-H2B interaction with HIRIP3?

Figure 3 panel B, HIRIP3-1-111 should be H2A-1-111.

In the conclusion section, I wouldn’t conclude HIRIP3 as a histone chaperone specifically for H2A, because it clearly has a CHZ domain that interact with a H2B.

In line 37, I do not agree with this statement, please provide a reference you think this is true.

In line 385, the authors claim that CK2 phosphorylate H2A within eHIRIP3 complex, which is misleading. The CK2 H2A phosphorylation experiment in vitro is informative, but this has nothing to do with eHIRIP3 complex. Please rephrase the whole section to reflect the real results.

In line 487, I didn’t get any conclusion based on the data available in the manuscript, so please provide more evidence as discussed above to show how HIRIP3 binds to H2A C alpha region.

Comments on the Quality of English Language

English is fine

Author Response

We thank this reviewer for his/her constructive and helpful suggestions. Below are presented our point-to-point responses to the reviewer’s comments and critical notes.

Minor error:

Previous works have shown that HIRIP3 binds to H2B, would this explain the minimal difference between H2A-H2B vs H2A.Z-H2B interaction with HIRIP3?

A>) This is a pertinent remark however the interaction with H2B has been ruled out by several other genetic structural studies (see reference 11, 12, 15 and 16). While H2B might not contribute to specific recognition by HIRIP3 in the same manner as H2A, it may function as a scaffold for chaperone-histone binding. The alphaC region of H2B could provide an extensive binding surface for the CHZ motif. This point is discussed in the discussion section.

Figure 3 panel B, HIRIP3-1-111 should be H2A-1-111.

A>) This was corrected in the new version of the manuscript

In the conclusion section, I wouldn’t conclude HIRIP3 as a histone chaperone specifically for H2A, because it clearly has a CHZ domain that interact with a H2B.

A>) The validation of the interaction with H2B could not be corroborated by other studies, including our own. Additionally, the structural analysis of the CHZ domain in interaction with H2AZ/H2B supports the conclusion that the CHZ domain is specific to H2A and does not engage with H2B.

In line 37, I do not agree with this statement, please provide a reference you think this is true.

In line 385, the authors claim that CK2 phosphorylate H2A within eHIRIP3 complex, which is misleading. The CK2 H2A phosphorylation experiment in vitro is informative, but this has nothing to do with eHIRIP3 complex. Please rephrase the whole section to reflect the real results.

A> We agree with the reviewer that this is an overstatement. We have now rephrased the whole section to reflect the real results. The title of the related paragraph was changed to “Recombinant CK2 phosphorylates H2A at serine 1”.

In line 487, I didn’t get any conclusion based on the data available in the manuscript, so please provide more evidence as discussed above to show how HIRIP3 binds to H2A C alpha region.

A> The deletion mutants illustrated in Figure 3B strongly indicate that residues 92-111 of H2A, encompassing the H2A alphaC region, play a crucial role in the interaction with HIRIP3. Another genetic study from the Kobor lab supports this finding, suggesting that Chz1 binds to all truncations except H2A.Z(1–104), which includes the alphaC region (Reference 11; DOI: 10.1128/MCB.05182-11). This point is now discussed in the discussion section.

Reviewer 3 Report

Comments and Suggestions for Authors

Identification and Characterization of HIRIP3 as a Histone H2A Chaperone. by Maria Ignatyeva et al.

This manuscript aims to analyze the structural basis by which HIRIP3 (homologue of yeast CHZ1) mediates preferential deposition of H2A.Z over H2A.  The authors use the HeLa cell model, which they subject to combination of approaches including:

1)     Isolation of protein complexes, identification of components by proteomics and MS (Fig. 1).

2)    Identify domains necessary for interaction of HIRIP3 with Histones H2A/H2A.Z/H2B and association with DNA (Fig. 2).

3)    Identify domains/motifs necessary for interaction of H2A with HIRIP3 (Fig. 3).

4)    Molecular modeling to evaluate similarity of yeast and human H2A/H2B (Fig. 4).

5)    Kinase assays to evaluate CK2-mediated phosphorylation of H2A and two site-directed mutations, Ser1 and Ser122 (Fig. 5).

Comments:

            The authors have made a concerted effort to better understand the potential role(s) of HIRIP3 in binding and loading of histones to DNA.  Unfortunately, there are several confounding issues which undermine the conclusions of the authors, and thus limit enthusiasm for the manuscript in its present form. 

            In Fig. 1D, the authors did not include a Western Blot for CC samples, and do not discuss why there is a discrepancy between the composition of CK2 in the SNC and CAC fractions.  Additionally, gels lack calibration markers, which are included in Fig. 1B.

            In Fig. 2B, pulldown assays do not show the input or unbound material in each analyzed sample.  In pulldown assays, it is advisable to show the Total (T, Input) as well as the unbound (S, supernatant) and bound (P, pellet) fractions to confirm that the ‘final pulldown signal’ reflects equal amounts of input materials.  These important controls are missing.  Similar problems affect Fig. 1C, D & E, which lack analysis of T-v-S-v-P fractions. 

In line 303, the authors state “Under these conditions, HIRIP3 lacking the CHZ domain exhibited significantly impaired deposition activity (Fig. 2F, lanes 3-5) compared to the full-length protein”.  However, this statement implies statistical analysis, which is not the case (it is a qualitative assessment) and cannot be made as there is no data to show that the two proteins in question (HIRIP3-FL and HIRIP3-∆CHZ) are at the same concentrations.  A more appropriate term would be ‘appears diminished compared to full-length’.  In line 306, the authors state “the CHZ domain is required”, a statement not supported by the data of Fig. 2F, as there is still some NCP loading with HIRIP3-∆CHZ.

            Fig. 5 is the most problematic.  Firstly, kinase assays are conducted at low ionic strength (see methods), at which CK2 activity is marginal, at best.  According to lines 206-208, 1 mg of each of the protein complexes were incubated for 30 minutes at 30°C in a buffer containing 70mM Tris-HCl, 10mM MgCl2, 5mM DTT, 0.5µ Ci P32 ATP and 100 ng of each of recombinant CK2α1/β and CK2α2/β.  It is difficult to reconcile these kinase assays with the known properties of CK2 to conclude that CK2 phosphorylates H2A within eHIRIP3 complex (see below).  The authors state that CK2 modifies Ser2 in the motif M1SGRGK--, which does not meet the consensus for CK2.  An extensive body of biochemical and molecular studies since the 1990s has established that:

1) CK2 is an acidophilic protein kinase.

2) Its consensus recognition sequence is S/T-(D/E)-x-(D/E), with strict requirements that basic residues such as Arg/Lys are excluded from positions n+1/2/3/4.  The presence of an Arg at n+2, a Lys at n+4 and the absence of Asp/Glu at n+1 and n+3, make it highly unlikely that this H2A is a target for CK2.  There is no evidence provided that the CK2 enzyme used in the studies is not contaminated with another ‘basophilic’ kinase, nor do the authors use a CK2-specific inhibitor to demonstrate that this non-canonical site in H2A is a bona fide target for CK2.  At the very least, the authors must clarify this issue with specific inhibitors.

On a separate note, it would seem more likely that the presence of CK2 in the protein complex with HIRIP3 is to modify this chaperone itself.  An examination of HIRIP3 sequence reveal a protein rich in Ser and Asp/Glu motifs, 12 of which appear to be excellent targets for CK2.  In fact, HIRIP3-CK2 complexes have been identified and it has been shown that this protein  is heavily phosphorylated by CK2 (see Assrir, et al. "HIRIP3 is a nuclear phosphoprotein interacting with and phosphorylated by the serine-threonine kinase CK2", vol. 388, no. 4, 2007, pp. 391-398., https://doi.org/10.1515/BC.2007.045).

In summary, it is equally/more likely that it is a HIRIP3-CK2 complex that binds H2A-H2B, and that CK2’s role in this complex may be to modulate HIRIP3 chaperone functions, a conclusion at odds with that of this manuscript.  Thus, the central hypothesis of the manuscript and the authors conclusions are not supported by the data.

Author Response

We thank this reviewer for his/her constructive and helpful suggestions. Below are presented our point-to-point responses to the reviewer’s comments and critical notes.

            In Fig. 1D, the authors did not include a Western Blot for CC samples, and do not discuss why there is a discrepancy between the composition of CK2 in the SNC and CAC fractions.  Additionally, gels lack calibration markers, which are included in Fig. 1B.

> We have incorporated calibration markers for all blots and submitted the original, non-cropped blots to the journal during the submission process. We question the rationale behind including a Western blot for the CC complex, as all its components are visibly present in the gel and have been analyzed by mass spectrometry. Furthermore, it is not the primary focus of the manuscript, which is centered on analyzing the specificity of HIRIP3 towards H2A.

            In Fig. 2B, pulldown assays do not show the input or unbound material in each analyzed sample.  In pulldown assays, it is advisable to show the Total (T, Input) as well as the unbound (S, supernatant) and bound (P, pellet) fractions to confirm that the ‘final pulldown signal’ reflects equal amounts of input materials.  These important controls are missing.  Similar problems affect Fig. 1C, D & E, which lack analysis of T-v-S-v-P fractions. 

A > We have adhered to the reviewer's suggestion and incorporated blots of the input or unbound material in each analyzed sample. It is essential for the reviewer to recognize that the presented experiments are not typical pull-down assays; rather, they are co-expression experiments conducted in vivo and subsequently pulled down using GST or highly specific antibodies under stringent conditions (0.3 M NaCl). Consequently, the specificity of the interactions observed should not be a point of contention.

In line 303, the authors state “Under these conditions, HIRIP3 lacking the CHZ domain exhibited significantly impaired deposition activity (Fig. 2F, lanes 3-5) compared to the full-length protein”.  However, this statement implies statistical analysis, which is not the case (it is a qualitative assessment) and cannot be made as there is no data to show that the two proteins in question (HIRIP3-FL and HIRIP3-∆CHZ) are at the same concentrations.  A more appropriate term would be ‘appears diminished compared to full-length’.  In line 306, the authors state “the CHZ domain is required”, a statement not supported by the data of Fig. 2F, as there is still some NCP loading with HIRIP3-∆CHZ.

A > We have followed the reviewer's suggestion and used the term “diminished”.

            Fig. 5 is the most problematic.  Firstly, kinase assays are conducted at low ionic strength (see methods), at which CK2 activity is marginal, at best.  According to lines 206-208, 1 mg of each of the protein complexes were incubated for 30 minutes at 30°C in a buffer containing 70mM Tris-HCl, 10mM MgCl2, 5mM DTT, 0.5µ Ci P32 ATP and 100 ng of each of recombinant CK2α1/βand CK2α2/β.  It is difficult to reconcile these kinase assays with the known properties of CK2 to conclude that CK2 phosphorylates H2A within eHIRIP3 complex (see below).  The authors state that CK2 modifies Ser2 in the motif M1SGRGK--, which does not meet the consensus for CK2.  An extensive body of biochemical and molecular studies since the 1990s has established that:

1) CK2 is an acidophilic protein kinase.

2) Its consensus recognition sequence is S/T-(D/E)-x-(D/E), with strict requirements that basic residues such as Arg/Lys are excluded from positions n+1/2/3/4.  The presence of an Arg at n+2, a Lys at n+4 and the absence of Asp/Glu at n+1 and n+3, make it highly unlikely that this H2A is a target for CK2.  There is no evidence provided that the CK2 enzyme used in the studies is not contaminated with another ‘basophilic’ kinase, nor do the authors use a CK2-specific inhibitor to demonstrate that this non-canonical site in H2A is a bona fide target for CK2.  At the very least, the authors must clarify this issue with specific inhibitors.

On a separate note, it would seem more likely that the presence of CK2 in the protein complex with HIRIP3 is to modify this chaperone itself.  An examination of HIRIP3 sequence reveal a protein rich in Ser and Asp/Glu motifs, 12 of which appear to be excellent targets for CK2.  In fact, HIRIP3-CK2 complexes have been identified and it has been shown that this protein is heavily phosphorylated by CK2 (see Assrir, et al. "HIRIP3 is a nuclear phosphoprotein interacting with and phosphorylated by the serine-threonine kinase CK2", vol. 388, no. 4, 2007, pp. 391-398., https://doi.org/10.1515/BC.2007.045).

In summary, it is equally/more likely that it is a HIRIP3-CK2 complex that binds H2A-H2B, and that CK2’s role in this complex may be to modulate HIRIP3 chaperone functions, a conclusion at odds with that of this manuscript.  Thus, the central hypothesis of the manuscript and the authors conclusions are not supported by the data.

A> While we agree with the reviewer's observation that the CK2 consensus recognition sequence is typically S/T-(D/E)-x-(D/E) and that CK2 targets HIRIP3, as suggested by Lipinski’s lab (Assrir, et al, reference 14) and outlined in the manuscript's introduction and discussion, it is important to note that CK2 has been demonstrated to phosphorylate noncanonical sequences, as documented in reference 29. Despite this, the S1A mutation notably abolished H2A phosphorylation, underscoring the critical role of the Serine 1 site in this process, even though it is not a canonical site for CK2 kinase. The purity of the recombinant CK2 kinase provided by Carna Biosciences is beyond doubt, as it has undergone comprehensive scrutiny by both the company and our team.

The primary specificity determinant for CK2 is an acidic residue at position +3, but additional acidic residues at positions spanning from -2 to +7 (and potentially beyond) could also act as positive specificity determinants for CK2, as discussed in reference 28. Consistent with this, incubation of CK2 with silmitasertib (CX-4945), a CK2-specific inhibitor, completely eliminated H2A phosphorylation (refer to the new Fig. 5C). However, we acknowledge the possibility that HIRIP3 might be the main target of CK2, as suggested in reference 14 and this does not affect the conclusions of our manuscript.

Regarding the kinase assay conditions conducted at low ionic strength (70mM Tris-HCl, 10mM MgCl2, 5mM DTT, 0.5µCi P32 ATP, and 100 ng of each recombinant CK2α1/β and CK2α2/β). The salt was not included in the reaction mix because it was provided by the recombinant H2A/H2B dimers stored in 1 M NaCl and 50 mM Tris (see M&M). Introducing 2 µl of H2A/H2B at 1M NaCl into a 20 µl reaction volume results in 100 mM NaCl in the reaction. This has now been corrected and included in the reaction buffer.

The stable coexistence of HIRIP3, CK2, and H2A/H2B within one complex implies a functional connection between these factors that may be subject to regulation through phosphorylation. In this context, H2AS1 phosphorylation has been demonstrated to serve as a pre-deposition modification signaling for the proper dimerization and deposition of H2A-H2B, as highlighted in reference 30.

This point is now discussed in the result section and in the last paragraph of the discussion section.

Round 2

Reviewer 2 Report

Comments and Suggestions for Authors

I am glad that the authors corrected the minor points since last time. However, I would like to see new results based on my first comments about experiments to test how the surrounding CHZ regions is involved in the interaction with H2A (or H2A.Z), but the authors didn't do anything about it. There is some in vivo studies on that, the authors shall refer to them, and expand the section of HIRIP3 and H2A/B interaction comparison to put some insights into the potential mechanism for its preference binding of H2A over H2A.Z. 

Comments on the Quality of English Language

English is fine

Author Response

We apologize for not accurately addressing the reviewer's inquiry regarding the role of the extended Chz1 region in H2A specificity. It appears that the reviewer is referencing a study on rice Chz1 conducted by Du et al. (DOI: 10.1038/s41467-020-19586-z), which elucidates the interaction between OsChz1 and both H2A-H2B and H2A.Z-H2B dimers. According to the study, OsChz1 exhibits a higher affinity for canonical H2A compared to the variant H2A.Z, with an extended Chz1 domain (amino acids 338-471) identified as responsible for the enhanced binding affinity to H2A-H2B dimers.

While the observation that OsChz1 displays a stronger affinity for H2A is interesting, it's worth noting that this protein significantly differs from mammalian HIRIP3. Sequence alignment analysis did not reveal a similar extended Chz1 motif in human HIRIP3. We have now referenced this noteworthy study in our discussion and deliberated on its findings in the context of our research.

We believe that we have effectively addressed this concern and duly recognized the associated findings.

Reviewer 3 Report

Comments and Suggestions for Authors

The authors have fully addressed issues raised with the originally submitted manuscript.  All relevant controls and edits to the text have been made and are found to be acceptable.  The revised manuscript is significantly improved, makes a strong case for the role of HIRIP3 as a H2A chaperone and the targeting of the latter by protein kinase CK2.  The revised manuscript is therefore considered suitable for publication.

Author Response

We thank this review for supporting the publication of our manuscript.